# Identifying Essential Hub Genes and circRNA-Regulated ceRNA Networks in Hepatocellular Carcinoma

**DOI:** 10.3390/ijms26041408

**Published:** 2025-02-07

**Authors:** Xiaoqian Yu, Hao Xu, Yutao Xing, Dehui Sun, Dangdang Li, Jinming Shi, Guangchao Sui, Guangyue Li

**Affiliations:** 1College of Life Science, Northeast Forestry University, Harbin 150040, China; xqyu@nefu.edu.cn (X.Y.); hao001221@nefu.edu.cn (H.X.); xingyutao@nefu.edu.cn (Y.X.); sundehui123@nefu.edu.cn (D.S.); lidd@nefu.edu.cn (D.L.); jmshi@nefu.edu.cn (J.S.); 2Institute of Biology, Westlake Institute for Advanced Study, Hangzhou 310030, China

**Keywords:** hepatocellular carcinoma, ceRNA network, circular RNA, microRNA, hub gene

## Abstract

Competitive endogenous RNAs (ceRNAs) absorb microRNAs and subsequently promote corresponding mRNA and long noncoding RNA (lncRNA) expression, which may alter cancer cell malignancy. Thus, dissecting ceRNA networks may reveal novel targets in cancer therapies. In this study, we analyzed differentially expressed genes (DEGs) of mRNAs and lncRNAs, and differentially expressed microRNAs (DE-miRNAs) and circular RNAs (DE-circRNAs) extracted from high-throughput sequencing datasets of hepatocellular carcinoma patients. Based on these data, we identified 26 gene modules using weighted gene co-expression network analysis (WGCNA), of which 5 were associated with tumor differentiation. In these modules, 269 genes were identified by GO and KEGG enrichment and patient’s survival correlation analyses. Next, 40 DE-miRNAs, each of which potentially bound a pair of DE-circRNA and hub gene, were discovered. Together with 201 circRNAs and 24 hub genes potentially bound by these miRNAs, 1151 ceRNA networks were constructed. Among them, 75 ceRNA networks consisting of 24 circRNAs, 28 miRNAs and 17 hub genes showed a positive circRNA–hub gene correlation. For validation, we carried out experiments for 4 randomly selected circRNAs regulating 19 potential ceRNA networks and verified 5 of them. This study represents a powerful strategy to identify essential gene networks and provides insights into designing effective therapeutic strategies.

## 1. Introduction

Hepatocellular carcinoma (HCC), the most common type of primary liver cancer, accounts for more than 90% of all primary liver cancer cases [1] and is the leading cause of cancer-related deaths worldwide [1,2], with the highest prevalence in East Asia and Africa [3]. Due to the liver’s anatomic features, HCC at early stage cannot be detected by ultrasound, which makes the diagnosis of primary HCC very challenging, and about 60% of liver cancer patients are diagnosed at late stages of the disease [4]. Despite the progress in diagnosis and therapeutic approaches, including immunotherapy and targeted therapy, the prognosis for HCC remains grim, with a five-year survival rate below 19% [5,6]. This stems from the complex genetic and epigenetic mechanisms that govern HCC development and progression [7,8,9].

Weighted gene co-expression network analysis (WGCNA) is a systems biology approach that analyzes the correlation patterns of genes across a large set of samples. It can both construct gene networks to detect gene modules and determine the essential players or hub genes within the modules [10,11]. WGCNA has been increasingly used as a powerful approach to determine modules of highly correlative genes and extensively employed to identify essential regulatory genes, candidate biomarkers or potential therapeutic targets in different cancers [12,13,14].

Circular RNAs (CircRNAs) are a class of single-stranded RNA molecules with sizes from 100 to over 4000 nucleotides (nts) and average lengths between 200 and 800 nts [15,16]. CircRNAs are formed by backsplicing or other non-canonical splicing mechanisms of exons or introns of precursor mRNAs (pre-mRNAs) [17,18]. This results in the formation of covalently closed, continuous loops of RNA molecules lacking 5’ and 3’ termini, which are resistant to digestion by exonucleases and thus more stable than linear RNAs [19,20,21]. Most circRNAs are noncoding RNAs, but an increasing number of them have been reported to encode proteins [22]. Long noncoding RNAs (LncRNAs) are a class of single-stranded RNA molecules longer than 200 nucleotides. They possess various biological functions, but lack protein-coding capability. Accumulating evidence indicates that lncRNAs play crucial roles in normal development and oncogenesis [23]. MicroRNAs (miRNAs) regulate gene expression by targeting mRNAs. Each miRNA has a seed sequence at the first 2–8 nts of its 5’-end, which perfectly matches a sequence in the 3’-untranslated region (UTR) of a target mRNA. When miRNA binds target mRNA mainly through its seed sequence’s recognition to a corresponding miRNA response element (MRE) at target 3’-UTR, it recruits the RNA-induced silencing complex (RISC) to promote mRNA degradation or block the protein translation process, leading to reduced gene expression [24].

The concept of competitive endogenous RNA (ceRNA) networks emerged in recent years, which proposes a mechanism that long RNA molecules, including lncRNAs and circRNAs, competitively bind miRNAs, and subsequently interfere with miRNA-mediated mRNA degradation or translational inhibition [25,26,27]. Each miRNA can bind multiple mRNAs to suppress their expression and each mRNA also harbors multiple MREs [28]. A circRNA can act as a ceRNA to bind or sponge miRNAs, which indirectly regulates the levels of mRNAs and lncRNAs targeted by these miRNAs [29]. For example, circ_0101675 can sponge miR-607 to upregulate PDL1 expression, leading to increased proliferation, migration, invasion, angiogenesis and immune evasion of non-small cell lung cancer cells [30]. In breast cancer, circ_001783 absorbs miR-200c-3p to improve the expression of its target genes ZEB1, ZEB2 and ETS1, and subsequently promote tumor growth [31]. Recent studies have underscored the critical role of ceRNA networks in the development and progression of various cancers, including HCC [32,33]. These networks hold promise to identify diagnostic and prognostic biomarkers and potential therapeutic targets of HCC [34]. For example, lncRNA NEAT1 functions as a ceRNA to regulate ferroptosis through the miR-362-3p/MIOX axis. Therefore, increased NEAT1 expression can absorb miR-362-3p and enhance MIOX expression, leading to cancer cell ferroptosis [35]. In liver cancer, lncRNA HOXD-AS1 acts as a ceRNA of miR-130a-3p and its increased expression can elevate SOX4 levels targeted by miR-130a-3p, leading to cancer metastasis [36]. Linc00974 can also absorb miR-642 to upregulate KRT19 expression, which causes the activation of Notch and TGF-β pathways that promote cancer cell proliferation [37].

Numerous studies have demonstrated the tissue-specific expression of circRNAs, paving the way for targeted cancer therapies with minimal systemic effects [38,39]. CircRNAs, owing to their remarkable stability against ribonuclease-mediated degradation, hold an advantage over linear lncRNAs [17,40]. Genome-wide analyses have revealed that circRNAs exhibit higher sequence conservation and abundance than their corresponding linear mRNAs [40,41]. By adjusting circRNA expression, researchers can effectively modulate the expression of associated mRNAs [42,43]. Consequently, targeted drugs designed against circRNAs are expected to exhibit relatively low off-target effects and highly specific efficacies. Notably, circRNAs have been reported to play pivotal roles in cancer initiation, progression and drug resistance [44,45]. Furthermore, circRNAs can influence the tumor microenvironment through intercellular communication mediated by their abundance in exosomes and body fluids [46]. Therefore, circRNAs hold immense promise as valuable cancer biomarkers and therapeutic targets.

Despite extensive research on circRNAs as miRNA sponges, systematic understanding to link their levels to gene expression remains elusive. In this study, we constructed an endogenous competition network involving circRNAs, miRNAs and their target genes to examine the potential of modifying circRNA expression for liver cancer treatment. We screened 24 circRNAs and 17 key genes to establish a ceRNA network, and based on their relative weights, we selected 4 circRNAs for validation. Subsequently, we silenced these four circRNAs and assessed the alterations in their corresponding genes and miRNAs. The findings of reverse transcription-quantitative (RT-qPCR) revealed five ceRNA networks associated with two circRNAs.

## 2. Results

### 2.1. Identification of Differentially Expressed Genes

We obtained the expression profiles of genes (referring only to those transcribing mRNAs and lncRNAs, hereafter) and miRNAs in liver cancer patients from The Cancer Genome Atlas (TCGA)-LIHC dataset [47]. Next, we analyzed these HCC tissues (n = 371) and para-carcinoma tissues (n = 50) for differentially expressed genes (DEGs) and differentially expressed microRNAs (DE-miRNAs). As a result, we identified 4667 DEGs (3392 upregulated and 1275 downregulated) and 242 DE-miRNAs (209 upregulated and 33 downregulated) (Figure 1a,b). The circRNA expression profiles were obtained from two GEO datasets (GSE169391 [48] and GSE122482 [49]). A total of 883 differentially expressed circRNAs (DE-circRNAs) (283 upregulated and 600 downregulated) were discovered (Figure 1c). The strategy of analyzing these differentially expressed RNAs to construct ceRNA networks is presented in Figure 1d.

### 2.2. Identification of Modules Using the WGCNA

We used the WGCNA to analyze the expression profiles of a total of 4667 significantly altered genes from the above identified DEGs. After preprocessing, 345 samples were used to construct a hierarchical clustering tree. The appropriate soft threshold was selected according to R^2^ = 0.9, and finally, the soft threshold β = 3 was selected to establish the relationship matrix (Figure 2a,b). Next, the degree of connection of each gene in the network was calculated to obtain gene distribution (Figure 2c). log(k) and log[p(k)] showed a linear negative correlation when β = 3 (Figure 2d). Then, gene feature modules were established and a total of 26 modules were obtained (Figure 2e,f).

### 2.3. Identification of Tumor Differentiation Modules

In the analysis of correlation between modules and clinicopathological traits, the turquoise (R = 0.28, *p* < 0.0001), purple (R = 0.25, *p* < 0.0001), blue (R = 0.27, *p* < 0.0001) and magenta modules (R = 0.24, *p* < 0.0001) were positively correlated with neoplasm histologic grade, while the brown module (R = −0.23, *p* < 0.0001) showed a significantly negative association with this trait (Figure 3). To ensure the reliability of the identification results of tumor differentiation-related modules, these modules were re-identified by calculating the mean absolute gene significance (GS) value of the genes for tumor differentiation in each module. The turquoise, purple, blue, magenta and brown modules showed the largest mean absolute GS values in all modules (Figure 4a). Therefore, these five modules have strong correlation with tumor differentiation. Subsequently, we used these five modules to identify tumor differentiation-related hub genes.

### 2.4. Identification of Tumor Differentiation-Related Hub Genes

The module membership (MM) of a gene is used to define the association between its expression and the co-expression of genes in a specific module. The MM value of each gene in the module was calculated to identify hub genes. Correlation analysis was performed for the GS of tumor differentiation and gene MM value in each module. According to the degree of correlation, whether an MM value is closely related to tumor differentiation characteristics can be determined. The results showed that the GS of tumor differentiation was significantly correlated with the MM values of the genes in the turquoise module (R = 0.67, *p* < 0.0001, Figure 4b), purple module (R = 0.63, *p* < 0.0001, Figure 4c), blue module (R = 0.62, *p* < 0.0001, Figure 4d) and brown module (R = 0.64, *p* < 0.0001, Figure 4e), but not the magenta module (R = 0.16, *p* = 0.068, Figure 4f). Therefore, the turquoise, purple, blue and brown modules display positive correlations with tumor differentiation. Our data also demonstrated that MM values can reflect the degree of correlation between genes and tumor differentiation. Thus, we chose these four modules to pursue hub genes.

Hub genes were identified based on GS and MM values of tumor differentiation using the ‘network screening’ function, which can obtain a q-weighted value for a specific gene in the module. By setting a threshold of q-weighted value < 0.0001, we totally obtained 406 potential hub genes (Appendix A), which were distributed in the turquoise, purple, blue and brown modules with 250, 25, 84 and 47 genes, respectively. Next, the biological functions of these 406 hub genes were analyzed using the Gene Ontology (GO) and Kyoto Encyclopedia of Genes and Genomes (KEGG) pathway enrichment analyses. The results showed that tumor differentiation-related hub genes were enriched in the cell cycle and DNA replication-related pathway (Figure 5a,b, Appendix A).

To explore the association of these 406 hub genes with the overall survival of the patients in the TCGA-LIHC dataset, we employed the Kaplan–Meier curve and Cox Proportional Hazards Model analyses. Accordingly, 269 of these 406 hub genes were identified for their association with overall survival (*p* < 0.05, Appendix A). We plotted 16 genes showing the most pronounced negative correlation with the overall survival of HCC patients (Figure 6). These 269 selected genes were used in the subsequent ceRNA network construction.

### 2.5. Construction of ceRNA Networks

MiRNAs that could potentially target the 269 hub genes were analyzed by Tarbase [50] and TargetScan [51] databases, which predicted 1862 and 281 miRNAs, respectively. Among them, 114 miRNAs were included by both databases (Figure 7a), in which 40 miRNAs (34 upregulated and 6 downregulated) were also present in the DE-miRNAs (Figure 1b). Next, we analyzed these 40 miRNAs for their potential binding to the 883 DE-circRNAs (shown in Figure 1c) using the circBank database [52]. To construct ceRNA networks, we searched for “paired” DE-circRNAs and hub genes, each of which shared at least one binding element of a specific miRNA belonging to these 40 DE-miRNAs, and obtained 201 circRNAs and 24 hub genes that met this criterion. Using these 201 circRNAs, 40 miRNAs and 24 hub genes, we identified 1151 potential circRNA/miRNA/hub gene networks (Appendix A). In these analyses, the differential expression data of the hub genes and DE-circRNAs were obtained from additional hepatocellular carcinoma-related whole-transcriptome sequencing datasets available in the GEO database (GSE216613 [53], GSE125469 [54] and GSE128274 [55]). Among these 1151 ceRNA networks, 75 showed positive circRNA–hub gene correlation, which contained 24 circRNAs, 28 miRNAs and 17 hub genes (Figure 7b,c, Appendix A).

### 2.6. Differential Expression Analysis: circRNAs, miRNAs and Hub Genes

From the upregulated DE-circRNAs identified in our analyses, we randomly picked four circRNAs, including circ_0000943, circ_0008346, circ_0001345 and circ_0103672, corresponding to 19 potential ceRNA networks (Appendix A), for further analyses within ceRNA networks. These four circRNAs were significantly upregulated in the liver cancer samples in the datasets GSE169391 and GSE122482 (Figure 8a). Next, we analyzed the miRNA binding sites of these circRNAs, and selected eight of the most affected miRNAs, including miR-424-5p, miR-103a-3p, miR-17-5p, miR-20a-5p, miR-20b-5p, miR-93-5p, miR-519d-3p and miR-195-5p.

Further investigation into the established ceRNA networks allowed us to pick six hub genes associated with these selected circRNAs and miRNAs, including CDC25A, CDCA4, CHEK1, CHAF1A, KPNA2 and OIP5. The activities of these genes are summarized in Table 1. We designed gene-specific primers (Appendix A), and carried out RT-qPCR. The upregulation of these circRNAs and hub genes in liver cancer HepG2 and Hep3B cells were verified (Figure 8b,c). However, the expression of the 8 selected miRNAs showed diverged patterns in these HCC cells; miR-103a-3p, miR-17-5p, miR-20a-5p, miR-20b-5p and miR-93-5p were upregulated, and miR-424-5p, miR-519d-3p and miR-195-5p were downregulated (Figure 8d).

### 2.7. Functional Evaluation of Identified circRNAs in HCC Cells

To investigate the importance of the 4 circRNAs (circ_0008346, circ_0000943, circ_0001345 and circ_0103672) in regulating HCC cell proliferation, we designed antisense oligonucleotides (ASOs) that targeted their circulating junction sites to eliminate their possible interference with the expression of corresponding linear mRNAs or lncRNAs (ASO sequences are available in Appendix A). We individually knocked down these circRNAs in HepG2 and Hep3B cells. Compared to the control group, the depletion of any of the four circRNAs led to significantly reduced viability and motility of HCC cells (Figure 8e,f, Appendix A). We also checked the effects of these ASOs on the four cognate mRNAs in liver cancer cells from which the four circRNAs are, respectively, produced through backsplicing. In HepG2 cells, the knockdown of circRNAs, except circ_0000943, could reduce the expression of corresponding parental genes, while in Hep3B cells, the knockdown of circRNAs, except circ_0103672, reduced the expression of corresponding parental genes (Appendix A). The mechanisms underlying the corresponding mRNA expression changes caused by circRNA knockdown deserve future investigation.

### 2.8. Validation of the Predicted circRNA/miRNA/Hub Gene Networks

To validate the regulatory roles of circRNAs in the predicted ceRNA networks, we analyzed whether the knockdown of each circRNA by its specific ASO could upregulate its potentially sponged miRNA(s) and subsequently downregulate the hub genes targeted by these miRNA(s). After individually introducing the ASOs into HepG2 and Hep3B cells, we used RT-qPCR to determine the expression of the miRNAs and hub genes in these networks, with graphic results shown in Figure 9 and altered gene expression summarized in Table 2.

First, in response to the circ_0008346 knockdown, miR-242-5p levels did not exhibit significant alteration, and the expression of CDC25A, CDCA4 and CHEK1 showed conflicting changes between HepG2 and Hep3B cell lines (Figure 9a,b). These results did not support the existence of the predicted ceRNA networks mediated by circ_0008346. Second, with circ_0000943 knockdown, miR-103a-3p showed opposite changes in the two cell lines, and its potential target OIP5 was upregulated (Figure 2c,d, left bar sets); meanwhile, miR-20b-5p exhibited a marked reduction in both cell lines, but its potential target KPNA2 was also downregulated (Figure 2c,d, right bar sets). These results did not validate the predicted ceRNA network regulated by circ_0000943. Third, when circ_0001345 was silenced, we observed increased levels of miR-17-5p, miR-20a-5p, miR-93-5p and miR-519d-3p, but not miR-20b-5p (Figure 9e). Both CHAF1A and KPNA2 harbor the potential target sites of all these 5 miRNAs in their 3’-UTRs. However, we only detected the downregulation of KPNA2 in the two cell lines, but CHAF1A showed opposite changes between them (Figure 9f). Thus, circ_0001345 likely serves as a ceRNA of miR-17-5p, miR-20a-5p, miR-93-5p and miR-519d-3p to upregulate KPNA2. Fourth, targeting circ_0103672 caused miR-103a-3p increase and OIP5 reduction (Figure 9g,h, left bar sets), suggesting a circ_0103672-mediated ceRNA network regulating OIP5 expression. However, this network does not perfectly apply to miR-195-5p and its potential target CDC25A, CDCA4 and CHEK1, only due to the reduced miR-195-5p level in Hep3B cells (Figure 9g,h, right one and three bar sets, respectively). In the alignment analysis, we confirmed the matching between the seed sequences of the five miRNAs with a target site (nts 102–108) in the 3’-UTR of the KPNA2 mRNA (Figure 9i). We also observed the inhibition of these five miRNAs in a reporter assay using a reporter vector with the KPNA2 mRNA 3’-UTR inserted downstream of the Gaussia luciferase (Gluc) coding sequence driven by the GSK promoter (Figure 9j). To further validate the circRNA-regulated network, we transfected HepG2 cells using the ASOs against circ_0000943 and circ_0001345 individually or simultaneously, and observed reduced KPNA2 protein expression in Western blot analysis (Figure 9k).

Overall, among the four selected circRNAs, we verify the five ceRNA networks of circ_0001345 and circ_0103672 in regulating KPNA2 and OIP5 expression, respectively (Figure 9i). Additionally, circ_0103672-regulated CDC25A, CDCA4 and CHEK1 expression through sponging miR-195-5p could only be validated in HepG2 cells (Table 2, bottom rows).

## 3. Discussion

Many previous studies demonstrated that altering the expression of genes or circRNAs within ceRNA networks through their competitive binding to shared miRNAs can impact the entire gene expression system and may impair cancer development [32,33,34]. Based on high-throughput sequencing datasets and bioinformatic analyses, we combined the hub genes identified by WGCNA with DE-miRNAs and DE-circRNAs. Using the circBank database, we constructed 1151 intricate ceRNA networks featuring regulatory interactions. Subsequently, we obtained 75 ceRNA networks with positive expression correlations between circRNAs and hub genes, indicating coordinated regulation within these modules. Focusing on 4 circRNAs regulating 19 ceRNA networks, we first demonstrated that these circRNAs played essential roles in maintaining the proliferation and migration of HCC cells, and then validated 5 circRNA-miRNA-mRNA networks (Figure 9i). Our finding sheds light on the crucial roles of circRNA-mediated ceRNA networks in liver cancer progression.

Silencing circ_0008346 and circ_0000943 effectively suppressed HCC cell progression, but the expression changes in their associated miRNAs and hub genes in the predicted ceRNA networks diverged from our prediction. Previous research indicated that circ_0008346 was upregulated in cancer. Interestingly, its corresponding mRNA, NUSAP1, is also increased in HCC, enhances the stemness of cancer cells, and promotes early cancer recurrence [64]. In addition, circ_000943 contains an m^6^A-modified start codon in its circulating junction site, which encodes a protein that promotes cancer progression by interacting with TFII-I proteins in the nucleus [65]. While our experimental data did not validate their involvement in the predicted ceRNA network, the oncogenic activities of circ_0008346 and circ_0000943 likely operate through other pathways or unidentified networks. Importantly, we validated that circ_0001345 acted as a ceRNA to upregulate KPNA2 expression through sponging four different miRNAs, including miR-17-5p, miR-20a-5p, miR-93-5p and miR-519d-3p. Meanwhile, our study also revealed that circ_0103672 mediated the ceRNA network of OIP5 through miR-103a-3p, thereby exerting oncogenic effects (Figure 9i).

It is noteworthy that we identified more hub genes as oncogenes than those as tumor suppressors. Based on the conception of cellular regulatory networks, most hub genes occupy the central positions in gene regulatory and/or protein–protein interaction networks, indicating that they have a high degree of connectivity and influence over many signaling pathways and subsequently lead to uncontrolled cell growth and proliferation, which share many characteristics with oncogenes. Certainly, tumor suppressor genes can also be situated at central positions and may be identified as hub genes, but with significantly smaller numbers due to their negative regulation in the most proliferative signaling pathways.

In the current study, when analyzing the correlation between modules and clinicopathological traits (Figure 3), we identified multiple genes with tumor suppressive activities using the TSGene 2.0 (https://bioinfo.uth.edu/TSGene/, accessed on 10 December 2024), including CHEK1, PLK1, MSH2, FANCG, RBL1 and BRCA2 (in the turquoise module), ESR1 (in the brown module) and GAS5 (in the blue module). To validate the ceRNA networks, we selected six genes, including CHEK1, based on their central roles and strong clinical correlations (Figure 6). When constructing the ceRNA networks, we observed that some of these genes, including CHEK1, RBL1 and ESR1, were involved in ceRNA interactions (Figure 7c).

Each cell is a complex entity with numerous interactive regulatory networks. Any change in genes, especially these hub genes or the ones at intersections, may inevitably impact many others. Bioinformatic analyses based on high-throughput data are unable to precisely illustrate the detailed landscape of the complicated regulatory networks in living cells, but may only provide us clues to strategize our experimental design. Therefore, it is unsurprising that we could only validate 5 ceRNA networks among 19 predicted ones. The intricate interplay, either direct or mostly indirect, among different genes or networks may contribute to this phenomenon. Meanwhile, failure in validating a predicted network may not necessarily exclude its existence, and the regulatory network could be submersed by many other cellular events and did not manifest itself enough to be detectable. Certainly, it may also suggest its insignificance in the entire network.

The major significance of the current study is the rational dissection of complex regulatory networks in cells and identification of the most possible functional interactions among different RNA molecules. Based on previous reports, up to 93% of the human genome can be transcribed [66]. A single mammalian cell may have 0.5–1.0 million of mRNAs encoded by only 1–5% of the genome [67], let alone other types of RNA molecules transcribed by the rest (around 90%) of the genome, including miRNAs (about 2300 types [68]), lncRNAs (encoded by over 50,000 genes [69]), and circRNAs (over 7000 types [16]), among many others. Most of them have various splicing isoforms expressed in different levels. Additionally, a eukaryotic cell contains about 42 million protein molecules [70]. Among this enormous number of molecules, it is impossible to precisely portray the cellular regulatory landscape, and any researcher would feel puzzled to start their experimental exploration to dissect functional interactions among RNA molecules. In this study, we presented a logical strategy that utilizes a combination of bioinformatic approaches to analyze the datasets of various RNA expression profiles derived from a particular cancer type. This can provide researchers with clues to strategize experimental designs when any of their concerned circRNAs, lncRNAs or microRNAs are present in a predicted ceRNA network. Importantly, as a system biology approach, the WGCNA, employed in our study, has been widely used to determine gene expression correlation patterns, detect gene modules and/or predict essential or hub genes in thousands of research papers since it was initially developed by Langfelder et al. in 2008 [10]. The extensive application demonstrates WGCNA as a powerful approach for gene network interpretation. Certainly, the current study is unable to precisely illustrate the detailed landscape of the complex regulatory networks in liver cancer cells, but we can map or predict the most possible gene expression context that will significantly increase the success rates of further experimental exploration.

In our validation experiments, we obtained some results contrasting the regulation of predicted ceRNA networks. For example, as shown in Table 2, circ_0000943 knock-down caused miR-20b-5p downregulation, although it still led to a predicted KPNA2 decrease. Circ_0103672 knockdown increased miR-195-5p expression in HepG2 cells, but reduced its levels in Hep3B cells, although all three potential target genes CDC25A, CDCA4 and CHEK1 of miR-195-5p were downregulated. Multiple reasons could lead to these observations. First, the two cell lines have different genetic and epigenetic backgrounds. For example, Hep3B cells harbor a functionally mutated p53, while HepG2 cells contain wild-type p53, which may cause a significantly different response to circRNA knockdown between the two cell lines. Certainly, additional genetic and epigenetic differences could also contribute to the observations. Second, a sponge circRNA can sequester miRNAs to prevent their binding to target mRNAs, but may not always reduce miRNA expression. As reported by Wightman et al., circRNAs may either enhance or inhibit target RNA-directed miRNA degradation [71], suggesting that sequestration by a circRNA does not necessarily lead to miRNA reduction. Third, an additional regulatory network may coexist. For example, circ-Amotl1 binds MYC to improve its stability and upregulate the expression of its target genes [72]. Meanwhile, MYC regulates the expression of various miRNAs, such as activating oncogenic miR-9, miR-17, miR-18a and miR-19a [73]. Thus, the intrinsic cellular difference and entangled regulatory circuits may affect the accuracy of the ceRNA network prediction.

The current study has clear restrictions. First, to specifically knock down a circRNA, we must design an ASO that targeted its circulating junction site to avoid disturbing the expression of its cognate mRNA or lncRNA. However, not all of these junction sites can serve as ideal targets for knockdown experiments. Therefore, different ASOs may have distinct inhibitory efficiencies, and certain circRNAs could not be well silenced when we validated their ceRNA activities. Second, when constructing a ceRNA network, we selected a circRNA that shared at least one miRNA binding element with its corresponding hub gene. However, when ciRS-7 was first discovered as a miRNA sponge in 2013, this circRNA was reported to contain over 70 conserved miR-7 target sites [74]. Therefore, it would be more informative and helpful to determine the number or enrichment of an miRNA-binding element in each circRNA. Third, different cancer cell lines have distinct genetic and epigenetic backgrounds, which may impact the execution of a predicted ceRNA network. Therefore, network validation in various hepatoma cell lines with consideration of their genetic alteration may provide more reliable results.

In the current study, we not only revealed many potential circRNA-mediated ceRNA networks in hepatocellular carcinoma, but also demonstrated a logical and strategical plan to explore essential regulatory networks in different cancers. Future research may be directed to explore the specific mechanisms and functions of identified circRNAs, miRNAs and novel hub genes in liver cancer, and develop circRNA-based diagnostic and therapeutic approaches. In addition, the circRNAs discovered in this study may also contribute to the development of other cancers, which certainly deserves future exploration.

Understanding of ceRNA networks can also provide insights into targeted cancer therapies. RNA molecules are more vulnerable than proteins for intervention. Especially, various therapeutic oligonucleotides, including ASOs, interfering RNAs, nucleic acid aptamers and DNAzymes, have been designed with great potential in cancer treatments [75]. Taking into account the high conservation and tissue specificity of circRNAs, we can advance cancer therapies by designing cancer-specific strategies to target essential regulatory mRNAs, circRNAs or miRNAs with minimized side effects.

## 4. Materials and Methods

### 4.1. Data Collection

The level 3 gene and miRNA expression data and the clinical information were obtained from a TCGA LIHC cohort including 371 primary tumor and 50 para-carcinoma tissues [47]. The circRNA and whole-transcriptome sequencing datasets were downloaded from the GEO database, including three whole-transcriptome sequencing datasets (GSE216613 [53], GSE125469 [54] and GSE128274 [55]) and two circRNA profiling datasets (GSE169391 [48] and GSE122482 [49]) (Appendix A).

### 4.2. Identification of DEGs, DE-miRNAs and DE-circRNAs

circRNA raw data (sra data) were downloaded from GEO and transformed to fastq data using a fasterq-dump program from the SRA-Toolkit 2.11.3, and then were passed for all quality checks using fastp v0.23.2 [76]. Passed reads were aligned to the UCSC human genome reference (hg 38) using HISAT2 2.2.1 [77]. Next, the circRNA identification and differential expression analysis were performed using CIRIquant v1.2.1 [78]. DEGs and DE-miRNAs were identified using R package DEseq2 1.40.2 [79], edgeR 3.42.4 [80] and the Wilcoxon test [81]. The final result used |log_2_FC| > 1 and FDR < 0.05 as a threshold to screen DEGs, DE-miRNAs and DE-circRNAs.

### 4.3. Co-Expression Network Construction

The DEGs screened from the TCGA-LIHC datasets were applied to establish the co-expression network construction using R package WGCNA 1.72.1 [10], according to a previously reported method [82], with minor modifications. The co-expression similarity matrix was transformed into an adjacency matrix by opting for a power of β = 3 as the soft threshold parameter to construct an unsigned scale-free network. Then, a topological matrix was constructed using the TOM.

For the classification of genes with similar expression patterns into gene modules, a TOM-based dynamic tree cut method was employed with the main parameters as follows: minModuleSize of 30 and deepSplit of 2. Finally, a cutting height (0.25) was selected for the module dendrogram, but no modules were merged depending on the estimated modular eigengenes (MEs), which were considered to be representative of gene expression profiles in the modules. The correlations between MEs and clinicopathological traits were calculated. A tumor sample with higher similarity to normal tissue suggested its lower histological grade or well differentiation, while the lower similarity of a sample to normal tissue indicated its advanced grade or poor differentiation with a highly proliferative and/or metastatic status. Thus, tumor differentiation indicates the ability of tumor proliferation and migration. Categorized MEs with correlation coefficients |R| > 0.2 and corresponding features of *p* values < 0.05 were selected for subsequent analysis (Figure 3).

### 4.4. Identification of Hub Genes and GO/KEGG Enrichment Analyses

The network screening function of the WGCNA was used to directly identify hub genes based on gene significance (GS, indicating the correlation between the gene and a particular clinical trait) and module membership (MM, representing the correlation between a gene and a specific module). A q-weight cutoff < 0.0001 was employed to obtain the hub genes. GO and KEGG enrichment analyses (over-representation analysis, ORA) were carried out using the R package clusterProfiler 4.8.1 [83], and the adjusted *p* value < 0.05 was considered significant.

### 4.5. Survival Analyses

The patients’ survival data extracted from the TCGA-LIHC clinical file and gene expression results were corrected using the Trimmed Mean of M-values (TMM) normalization method available in the edgeR package 3.42.4 [80]. These results were used as input to the Survival R package (v3.5.5) to categorize patients into high and low expression groups based on the median expression levels, calculate the *p* values of hub genes and plot the survival curves. Cox’s proportional hazards regression model was used to calculate hazard ratios (HRs). The *p* values < 0.05 were considered significant, and 269 of 406 hub genes were used in subsequently analyses.

### 4.6. Procedure to Construct ceRNA Networks

The potential miRNAs regulating selected hub genes were predicted by Tarbse 9.0 [50] and TargetScan 7.0 [51] databases. The potential mRNA-miRNA pairs that were consistent across the two datasets, and the circRNA-miRNA pairs that were predicted using the circBank 1.0 [52] database, were merged to establish ceRNA networks. Then, we used other whole-transcriptome sequencing data to filter the networks. The networks with a positive correlation between protein-coding genes and circRNAs were retained.

### 4.7. Reverse Transcription-Quantitative PCR Analyses

Total RNA was extracted from cultured cells using the TRIzol reagent (Thermo Fisher Scientific Inc., Shanghai, China). Next, cDNA synthesis was carried out using M-MLV reverse transcriptase (Vazyme Biotech Co., Ltd., Nanjing, China). For the quantification of circRNA and mRNA levels, 2 μg of total RNA was incubated with 1 μg/μL of random hexamer primer at 65 °C for 5 min, followed by 4 °C for 2 min. Reverse transcription was conducted at 42 °C for 30 min and the sample was then chilled at 4 °C. Next, quantitative PCR (qPCR) was then performed using gene-specific primers and the Light Cycler 480 SYBR Green PCR Master Mix (Roche, Basel, Switzerland) on a Light Cycler 480 instrument (Roche, Basel, Switzerland). The qPCR conditions were as follows: 95 °C for 3 min, followed by 40 cycles of 95 °C for 15 s and 60 °C for 1 min. All reactions were performed in triplicate. The results were analyzed using the 2^−ΔΔCt^ method and normalized against β-actin. For mature miRNA quantification, stem-loop RT-PCR was performed as previously described [84], with normalization against U6. The primers used for the stem-loop RT-PCR are shown in Appendix A.

### 4.8. Cell Culture, Transfection and Western Blot

HL-7702, HepG2 and Hep3B cells were obtained from the American Type Culture Collection (ATCC), and cultured following the protocols of the vendor, using culture media from Gibco and fetal bovine serum (FBS) from ExCell Bio. ASOs against circRNAs and miRNA mimics were synthesized by Genewiz, Inc. (Suzhou, China) Western blot analysis was carried out using primary antibodies against KPNA2 (cat# RPE748Hu01, Cloud-Clone Corp., Wuhan, China) and β-actin (cat# A5441, Sigma-Aldrich, Shanghai, China). Lipofectamine 2000 (Thermo Fisher Scientific, Shanghai, China) was used to transfect cells with 150 nM of each ASO or 50 nM of each miRNA mimic following the manufacturer’s instructions.

### 4.9. Cell Viability, Wound Healing and Reporter Assays

Cell viability and wound healing assays were performed as previously reported [85]. In these experiments, cells were transfected by 150 nM ASO. In reporter assays, the 3’-UTR of human *KPNA2* mRNA was subcloned downstream of the *Gluc* coding sequence driven by the phosphoglycerate kinase (*PGK*) promoter, leading to the construction of the reporter vector *PGK*promoter-*Gluc*-*KPNA2*-3’-UTR. One µg of the reporter vector was cotransfected with 50 nM of each miRNA mimic or a control, and 50 ng of an expression plasmid of the secreted alkaline phosphatase (*SEAP*) into HeLa cells cultured in 12-well plates. The aliquots of the medium from transfected wells were collected at the 48 h time point after transfection to measure Gluc activity using coelenterazine (CTZ) as the substrate, and the data were normalized with the SEAP expression. Each experiment was performed in triplicate.

### 4.10. Statistical Analysis

All statistical analyses were performed by GraphPad Prism 8.3.0. The results were presented as a mean with either standard deviation (SD) or a standard error of mean (SEM). Two groups were analyzed by a two-tailed Student’s *t*-test. Statistical significance levels were denoted as follows: ns: no significance, *p* > 0.05, * *p* < 0.05, ** *p* < 0.01, *** *p* < 0.001, **** *p* < 0.001.

## Figures and Tables

**Figure 1 ijms-26-01408-f001:**
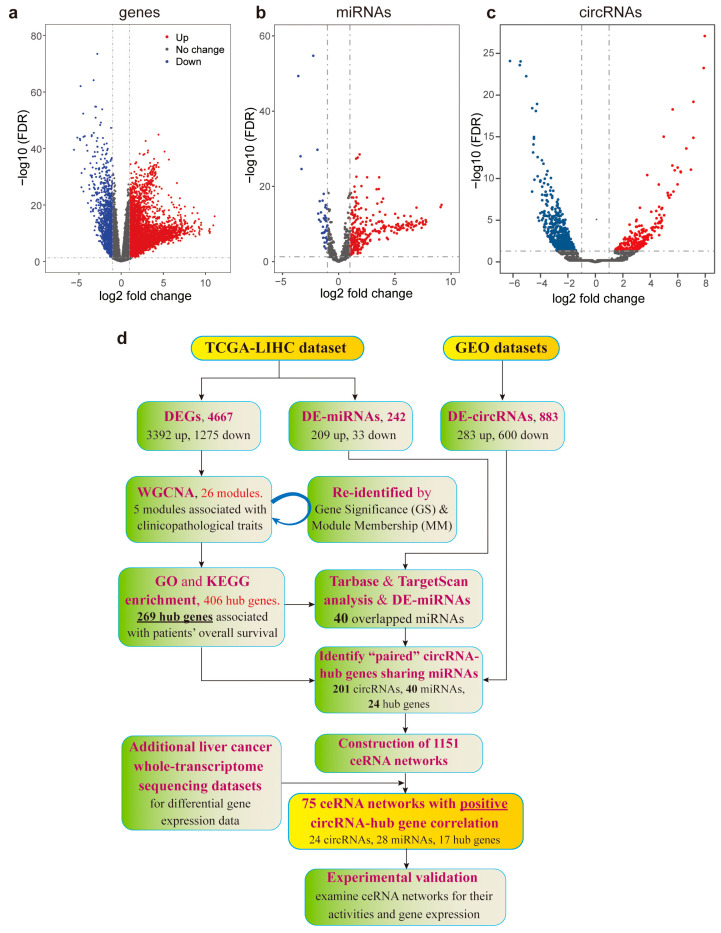
Differential expression analyses of RNAs in hepatoma carcinoma and strategy for their ceRNA network study. (**a**–**c**) Volcano plots of differentially expressed mRNAs and lncRNAs (**a**), miRNAs (**b**) and circRNAs (**c**) in the TCGA-LIHC dataset and two GEO datasets (GSE122482 and GSE169391). (**d**) Analytic strategy of differentially expressed RNAs to construct ceRNA networks.

**Figure 2 ijms-26-01408-f002:**
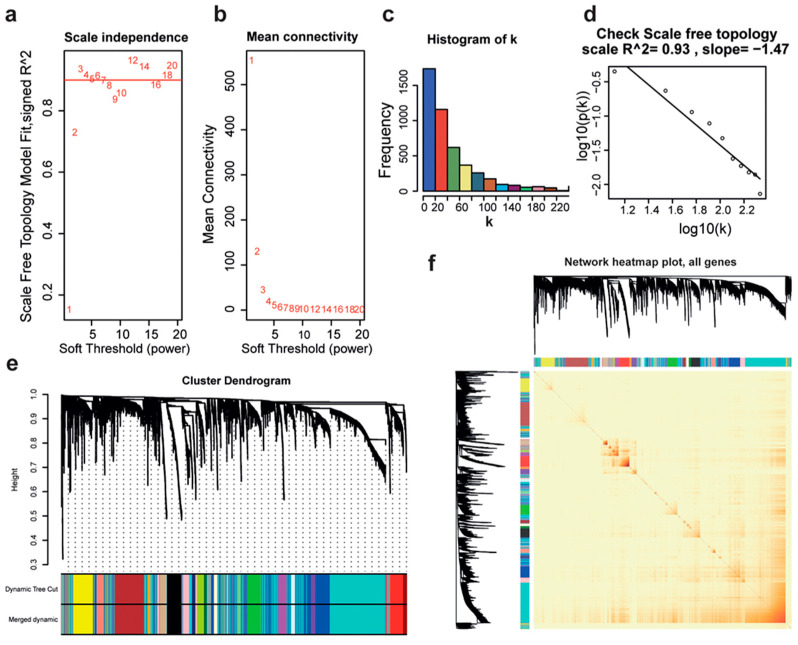
Construction of co-expression networks using the WGCNA. (**a**) The analysis of a scale-free topology fit index (R^2^ = 0.9) for various soft-thresholding powers (weighting coefficient, β). (**b**) Mean connectivity under different weighting coefficients. In (**a**,**b**), the numbers (ranging from 1 to 20) represent the soft-thresholding power to create the gene co-expression correlation matrix and achieve a scale-free network. Higher powers result in fewer but stronger connections in the network. (**c**) The histogram of the gene connectivity distribution in the co-expression network, with the degree of connection, k. (**d**) The plot of the scale-free topology in the network, and correlation of log (k) and log [P(k)]. (**e**) Cluster analysis to detect the co-expression clusters with corresponding color assignments. Each color represents a module in the gene co-expression network constructed by WGCNA. (**f**) Heatmap describing the topological overlap measure (TOM) of genes selected for WGCNA. Light colors indicate lower overlap and red colors indicate higher overlap. In (**e**,**f**), each color represents a distinct module identified by WGCNA, with genes grouped based on their co-expression patterns. The grey module represents genes that were not assigned to any other specific module.

**Figure 3 ijms-26-01408-f003:**
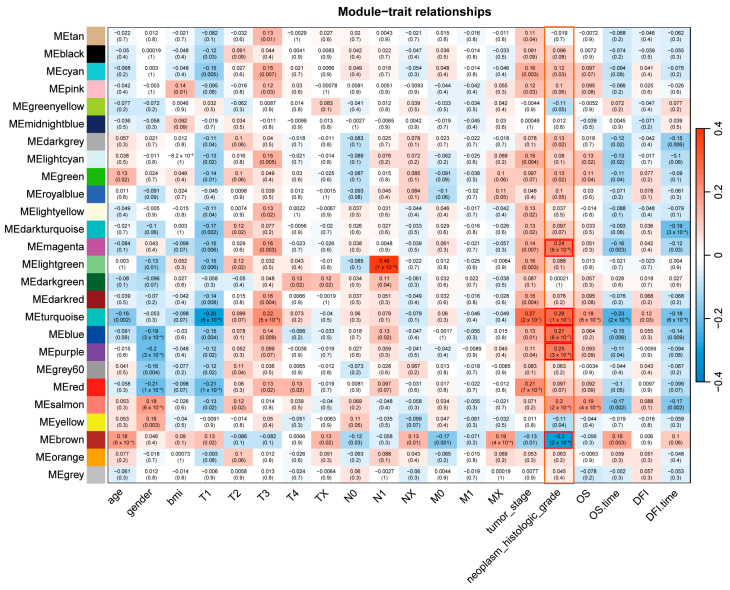
Correlation map between modules and clinicopathological traits. The 5 modules with significant positive or negative correlation to the clinicopathological traits are framed.

**Figure 4 ijms-26-01408-f004:**
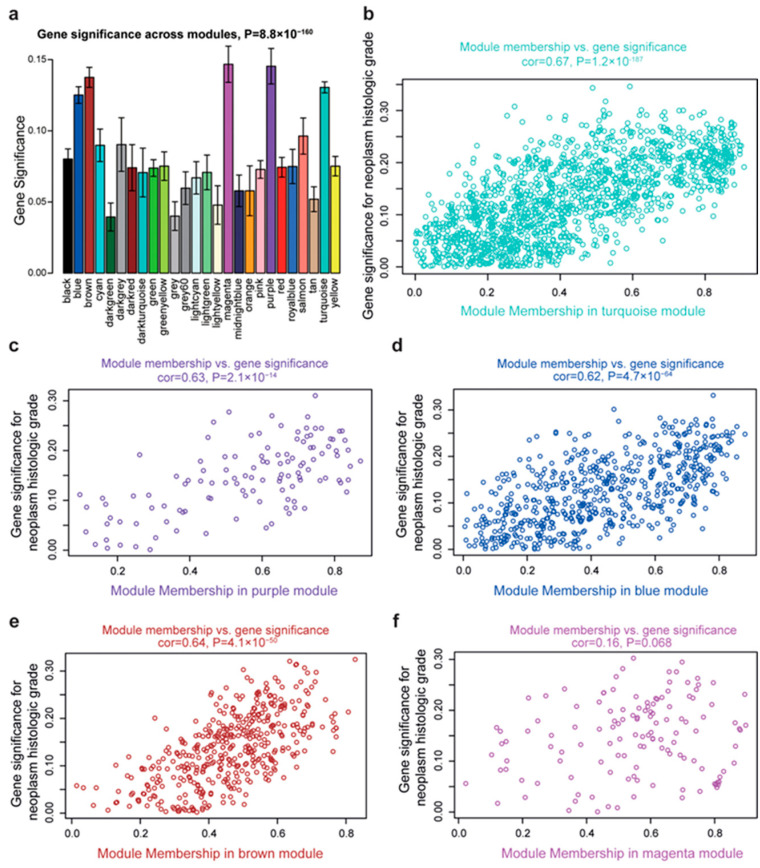
Identification of tumor differentiation-related modules and hub genes. (**a**) Average gene significance for these regulating tumor differentiation within each co-expression module. (**b**–**f**) Scatterplots of correlation analyses of MM (x-axis) versus gene significance for tumor differentiation (y-axis) in the five modules.

**Figure 5 ijms-26-01408-f005:**
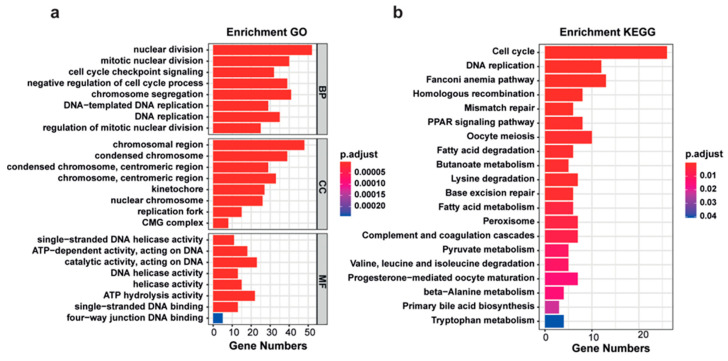
Enrichment analyses of tumor differentiation-related hub genes. (**a**,**b**) GO (**a**) and KEGG (**b**) enrichment analyses of 406 hub genes for their gene functions and relevant signaling pathways, respectively.

**Figure 6 ijms-26-01408-f006:**
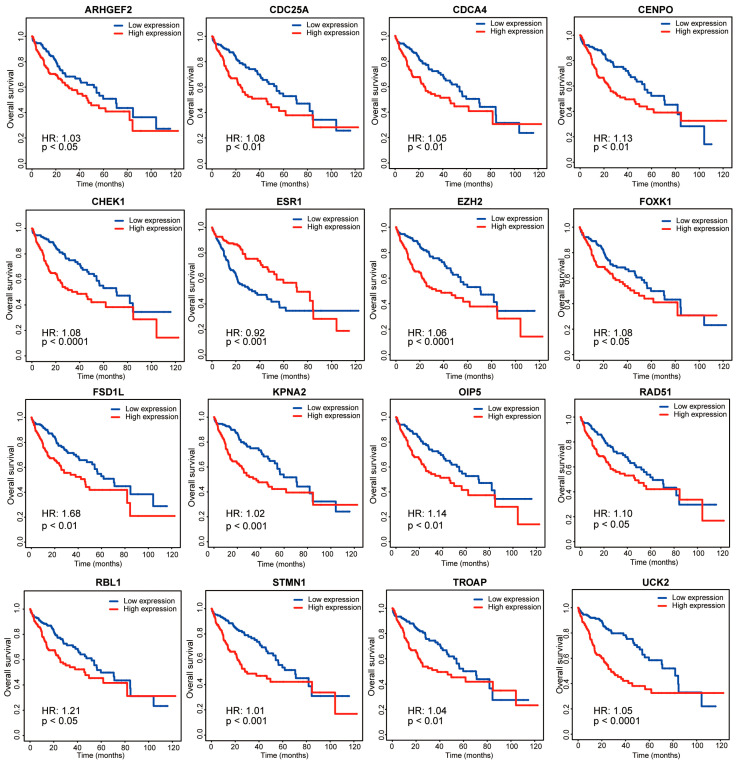
Kaplan–Meier curves of the association between 16 essential hub genes and overall survival of the patients from the TCGA-LIHC dataset. HR, hazard ratio.

**Figure 7 ijms-26-01408-f007:**
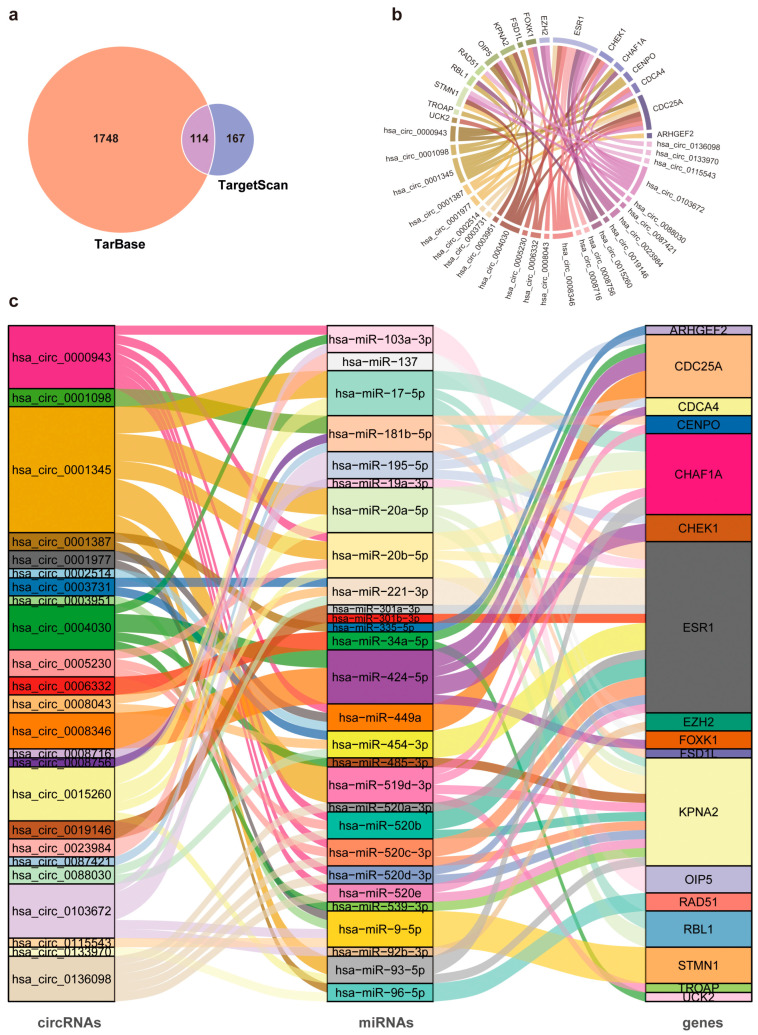
Construction of circRNA-mediated ceRNA networks. (**a**) Venn diagram showing the overlap of predicted miRNA-targeted survival-associated hub genes between TarBase and TargetScan databases. (**b**) GOChord plot displaying the correlation between circRNA and hub gene expression. (**c**) Schematic diagram showing internal connections in 75 predicted circRNA/miRNA/hub gene networks.

**Figure 8 ijms-26-01408-f008:**
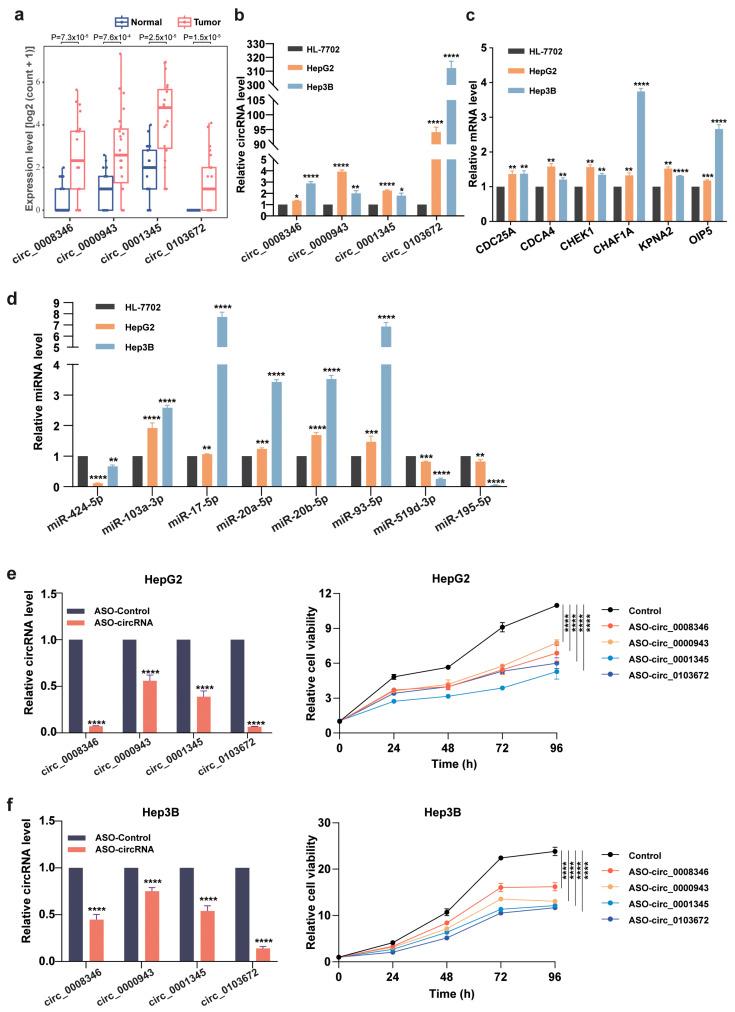
Determination of RNA expression by RT-PCRs and circRNA knockdown’s effects on cell viability. (**a**) Expression analysis of circ_0008346, circ_0000943, circ_0001345 and circ_0103672 in two GEO datasets (GSE169391 and GSE122482). (**b**–**d**) Quantification of circRNA, mRNA and miRNA levels by RT-qPCR in normal liver HL-7702 cells, HCC HepG2 cells and Hep3B cells. (**e**,**f**) Evaluation of ASO-mediated knockdown of 4 circRNAs (**left**) and its effects on the viability (**right**) of HepG2 (**e**) and Hep3B (**f**) cells. * *p* < 0.05, ** *p* < 0.01, *** *p* < 0.001, **** *p* < 0.001.

**Figure 9 ijms-26-01408-f009:**
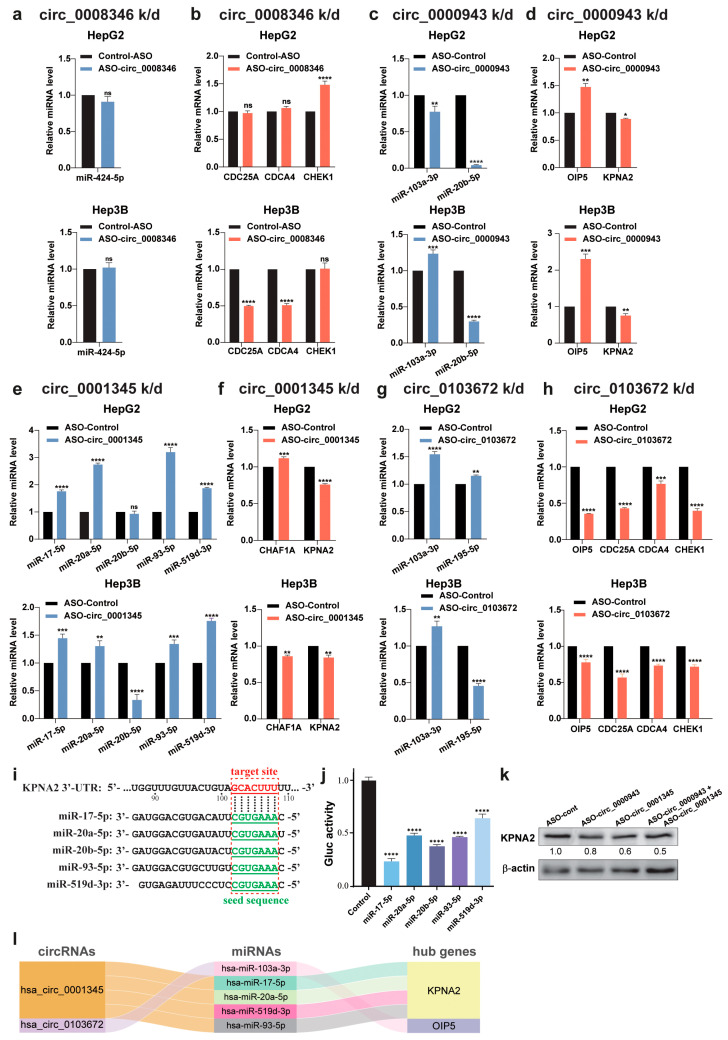
Validation of predicted ceRNA networks in HCC cells. (**a**–**h**) Effects of the knockdown of circ_0008346 (**a**,**b**), circ_0000943 (**c**,**d**), circ_0001345 (**e**,**f**) and circ_0103672 (**g**,**h**) on the expression of their potentially regulated miRNAs and mRNAs detected by RT-qPCR in HepG2 (**top** panels) and Hep3B (**bottom** panels) cells. (**i**) Diagram to align the seed sequences (green letters) of 5 miRNAs and the target sequence (nts 102–108, with the nt right after the stop codon designated as “1”) in the 3’-UTR of the KPNA2 mRNA. (**j**) Reporter assay to test the effects on the five miRNAs on the expression of Gaussia luciferase (Gluc) located upstream of the KPNA2 mRNA 3’-UTR. (**k**) Western blot analysis of KPNA2 expression in HepG2 cells transfected by the ASOs, as labeled on the top. The numbers under the KPNA2 blot are the results of densitometric analysis against corresponding β-actin bands. (**l**) Schematic diagram showing the 5 verified ceRNA networks. * *p* < 0.05, ** *p* < 0.01, *** *p* < 0.001, **** *p* < 0.001, ns: no significance.

**Table 1 ijms-26-01408-t001:** Hub genes in selected potential ceRNA networks and their biological activities.

Hub Genes	Cancer-Related Activities	Ref.
CDC25A (cell division cycle 25A)	As a phosphatase, it activates CDK2, CDK4 and CDK6 by removing their inhibitory phosphorylation. It also inhibits apoptosis signal-regulating kinase 1 (ASK1) to enhance cells’ resistance to apoptotic stimuli. Its overexpression promotes HCC tumor growth and is associated with poor prognosis of patients.	[56,57]
CDCA4 (cell division cycle-associated protein 4)	Its overexpression is observed in various cancers and correlates with poor clinical outcomes of cancer patients. Its knockdown markedly promotes HCC cell apoptosis, suggesting its role as an oncogene and potential therapeutic target.	[58]
CHEK1 (checkpoint kinase 1)	It is an essential regulator of DNA damage response and cell cycle checkpoint control to surveil the fidelity and integrity of the major cell cycle events. Its isoform CHK1-S (without Exon 3) acts as an oncogene and is upregulated in HCC. Overall, this gene is considered a therapeutic target in cancer therapies.	[59,60]
OIP5 (Opa interacting protein 5)	It is overexpressed in a variety of cancers. In HCC, its overexpression correlates with reduced survival rates. Its knockdown decreases cell proliferation and clonogenicity, leads to cell cycle arrest and promotes cell apoptosis.	[61]
KPNA2 (karyopherin subunit alpha 2)	It belongs to the nuclear transporter family and transports many cancer-related proteins. It is overexpressed in various cancers and associated with poor prognosis. It promotes cell proliferation, tumor formation and progression.	[62]
CHAF1A (chromatin assembly factor 1 subunit A)	It is a subunit of CAF-1, a protein complex regulating nucleosome formation by depositing H3/H4 onto DNA. Its overexpression is observed in various cancers, and is associated with poor prognosis, an immunosuppressive microenvironment and treatment resistance.	[63]

**Table 2 ijms-26-01408-t002:** Summary of experimental results with individual circRNA knockdown.

circRNAs	miRNAs	circRNA k/d *		Hub Genes	circRNA k/d *	
		HepG2 Cells	Hep3B Cells		HepG2 Cells	Hep3B Cells
hsa_circ_0008346	miR-424-5p	NS	NS	CDC25A	NS	Dn
				CDCA4	NS	Dn
				CHEK1	Up	NS
hsa_circ_0000943	miR-103a-3p	Dn	Up	OIP5	Up	Up
	miR-20b-5p	Dn	Dn	KPNA2	Dn	Dn
hsa_circ_0001345	miR-17-5p	Up	Up	CHAF1A	Up	Dn
	miR-20a-5p	Up	Up			
	miR-20b-5p	NS	Dn			
				KPNA2	Dn	Dn
	miR-93-5p	Up	Up			
	miR-519d-3p	Up	Up			
hsa_circ_0103672	miR-103a-3p	Up	Up	OIP5	Dn	Dn
	miR-195-5p	Up	Dn	CDC25A	Dn	Dn
				CDCA4	Dn	Dn
				CHEK1	Dn	Dn

* k/d: knockdown; NS: no significance; Up: upregulated; Dn: downregulated.

## Data Availability

Publicly available datasets were analyzed in this study. The following information was supplied regarding data availability: GEO (https://www.ncbi.nlm.nih.gov/geo/, accessed on 1 April 2023): GSE216613, GSE125469, GSE128274, GSE169391, GSE122482; TCGA database (https://portal.gdc.cancer.gov/, accessed on 26 March 2023); Tarbse (https://dianalab.e-ce.uth.gr/tarbasev9, accessed on 26 March 2023); TargetScan (https://www.targetscan.org/vert_80/, accessed on 26 March 2023); and circBank (http://www.circbank.cn/, accessed on 13 March 2023).

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
