# Peer review of "Identifying Essential Hub Genes and circRNA-Regulated ceRNA Networks in Hepatocellular Carcinoma"

_ijms, 2025, doi:10.3390/ijms26041408_

Round 1
Reviewer 1 Report (Previous Reviewer 1)
Comments and Suggestions for Authors
The manuscript by Yu et al. investigates the interaction between essential hub genes and circRNA-regulated ceRNA networks in hepatocellular carcinoma.
The authors have comprehensively revised their manuscript and have addressed all the previous comments. The manuscript can be accepted in its present form.
Author Response
Reviewer 1:
The manuscript by Yu et al. investigates the interaction between essential hub genes and circRNA-regulated ceRNA networks in hepatocellular carcinoma.
The authors have comprehensively revised their manuscript and have addressed all the previous comments. The manuscript can be accepted in its present form.
Reply: We thank the reviewer for the positive comments. The suggestions from the reviewer have helped us to improve the quality of the manuscript.
Reviewer 2 Report (New Reviewer)
Comments and Suggestions for Authors
The authors provide a detailed analysis of circRNA working as competitive endogenous RNA in hepatocellular carcinoma. They start by using WGCNA to find network fo circRNAs in patient data finally identifying 19 circ/ceRNAs. Their focus is on miRNA and circRNA interactions thus the introduction could improve if it is detailed about the known interactions of these kinds.
Could the authors check for some example circRNAs that their host genes are not downregulated? Also, the interaction between circRNA and miRNA is predicted. Is their any database or other information if these can be verified at the cellular level?
Could the authors please clarify “gene expression results were corrected by edgeR” at line 503. Are they saying this about the differentially expressed genes? Also, please provide the full form of the abbreviations used in the data showing cox regression.
What was the post-hoc test used for anova? The scratch assays are very hard to visualize for the effect the authors are claiming. If they have more time points between 0-72 that could be helpful or plotting as curve. Was the closure percentage for 72% only?
Line 72, should the reference be 30? Line 94, the authors could cite specific studies instead of two reviews to show how circRNAs can modulate the expression of their target RNA. Could the authors check the references for GSE122482. There is no reference listed on the geo site.
Author Response
Reviewer 2:
The authors provide a detailed analysis of circRNA working as competitive endogenous RNA in hepatocellular carcinoma. They start by using WGCNA to find network of circRNAs in patient data finally identifying 19 circ/ceRNAs. Their focus is on miRNA and circRNA interactions thus the introduction could improve if it is detailed about the known interactions of these kinds.
Reply: We thank the reviewer for the comments. In the Introduction section of this manuscript, we provided a brief summary of circRNA and miRNA interaction, especially the description and examples of circRNAs as ceRNAs to sponge and regulate miRNAs (starting from line 60). Subsequently, circRNAs may regulate mRNA expression through modulating the action of miRNAs (starting from line 87).
Could the authors check for some example circRNAs that their host genes are not downregulated? Also, the interaction between circRNA and miRNA is predicted. Is their any database or other information if these can be verified at the cellular level?
Reply: We thank the reviewer for the insightful comments. In our study, we checked the effects of circRNA knockdown on their host genes. As shown in Supplementary Figures 1c and 1d, knockdown of circRNAs generally reduced the expression of their host genes. However, knockdown of circ_0000943 did not significantly affect its host gene ARHGAP35 expression in HepG2 cells, while the knockdown of circ_0103672 did not change the expression of its host gene GATM in Hep3B cells. The different responses of host gene expression to circRNA knockdown can be attributed to the genetic and epigenetic backgrounds between the two cell lines. The underlying mechanisms causing these differences need future investigation.
For the prediction of circRNA and miRNA interaction, several databases are available, such as CircBank, CircInteractome and CircNet. However, different databases may use distinct algorithms to predict circRNA-miRNA interaction, and therefore may not always give consensus results. Especially, we found that some databases did not contain the circRNAs that we studied and thus cannot be used to analyze some circRNAs in this manuscript. Therefore, we focused on the outcome data of CircBank that is popularly used and has a very rich collection of circRNAs. Nevertheless, experimental validation is necessary for any future exploration based on the predicted circRNA and miRNA interaction.
Could the authors please clarify “gene expression results were corrected by edgeR” at line 503. Are they saying this about the differentially expressed genes? Also, please provide the full form of the abbreviations used in the data showing cox regression.
Reply: We thank the reviewer for pointing this out. We employed the Trimmed Mean of M-values (TMM) normalization method available in the edgeR package to adjust the raw count values and ensure accurate gene expression levels. This normalization process was applied to the dataset used to identify differentially expressed genes. Subsequently, we utilized the Kaplan-Meier method to generate overall survival Kaplan-Meier curves based on the median expression level of each gene. In the revised manuscript, we have added the related description in the Materials and Methods section, which can be found in lines 501-503.
As suggested by the reviewer, we have updated and also simplified the Supplementary Table 4 to replace abbreviations related to Cox regression results by their full forms. Meanwhile, we also add the hazard ratio (HR) for each curve. These changes and addition ensure the clarity and facilitates a better understanding of the presented data.
What was the post-hoc test used for anova? The scratch assays are very hard to visualize for the effect the authors are claiming. If they have more time points between 0-72 that could be helpful or plotting as curve. Was the closure percentage for 72% only?
Reply: We thank the reviewer for the comments. Actually, we did not use ANOVA in the data analysis of this manuscript. Our mention of ANOVA in the Materials and Methods section was an oversight and we apologize for this.
Our analysis did not involve multiple group comparisons that would necessitate the use of ANOVA and subsequent post-hoc testing. We have removed “Multiple groups were analyzed by two-way analysis of variance (ANOVA)” in the “4.10. Statistical analysis” of the revised manuscript.
Regarding the visualization of scratch assays, we calculated the data from three independent experiments and plotted the curves of closure percentage (presented in the right panels of Supplementary Figures 1a and 1b). We acknowledge that the error bars are generally large due to the nature of the scratch assays; nevertheless, we observed the significant difference between the control group and the four experimental groups at the 72 h time point.
Line 72, should the reference be 30? Line 94, the authors could cite specific studies instead of two reviews to show how circRNAs can modulate the expression of their target RNA. Could the authors check the references for GSE122482. There is no reference listed on the geo site.
Reply: We thank the reviewer for the comments. We checked the references indicated by the reviewer. (1) In line 72, we confirm that the reference should be #29. This reference is a book chapter (doi.org/10.1016/B978-0-443-14008-2.00002-4) discussing different forms of coding and noncoding RNAs in exosomes, and contains the concept of circRNAs that can sponge miRNAs to regulate the levels of mRNAs and lncRNAs targeted by the miRNAs. (2) When discussing “By adjusting circRNA expression, researchers can effectively modulate the expression of associated mRNAs”, we cited two experimental studies as references (PMID: 30053867 and 32193155), and removed the previous two references. (3) We have checked the reference for the GSE122482 dataset and confirmed the reference as PMID: 30537115. In the revised manuscript, this reference is #49 (it was #47 previously). The sample numbers, authors’ names and authors affiliations in this reference were consistent with those in the GSE122482 dataset.
This manuscript is a resubmission of an earlier submission. The following is a list of the peer review reports and author responses from that submission.
Round 1
Reviewer 1 Report
Comments and Suggestions for Authors
The manuscript by Yu et al. investigates the interaction between essential hub genes and circRNA-regulated ceRNA networks in hepatocellular carcinoma.
I have a few suggestions for polishing the commentary.
Comments
1. The authors should state clearly the novelty of their study in a clearer manner. They should highlight how the present study could be an interesting avenue for drug delivery applications.
2. Sponging actions of the ncRNAs is a very important part of this review. The authors should cite the following articles
https://doi.org/10.1111/acel.12484
https://doi.org/10.1016/B978-0-443-14008-2.00002-4
https://doi.org/10.1038/s41419-018-1287-1
https://doi.org/10.1007/s10528-023-10493-8
or any other relevant articles to validate this point.
3. The authors got some contrasting results with respect to some of the predicted ceRNA networks mediated by the circular RNAs. Is there any epigenetic regulatory role that is playing a role in it? The authors should clearly state this possibility in a detailed manner.
4. The authors have heavily focused on the negative correlation between the ceRNA network and the miRNA population. Is there any possibility of the positive correlation between ceRNA network and the miRNA or mRNA population that is responsible for the progression of HCC.
Reviewer 2 Report
Comments and Suggestions for Authors
The authors took several datasets to construct ceRNA network, and used the 5 candidate circRNAs for validating their hypothesis, including the circ_0001345. There are still some concerns awaited the authors to clarify.
Major
1. Figure 6 showing the highly correlation between hub genes and patient survival, but no tumor suppressor function hub genes were presented for its association with cancer malignancy. Whether the authors observed the tumor suppressor of hub genes in their analysis? If not, what is the probable reason for it?
2. The authors took 5 targets for evaluating the efficiency of ceRNA network. However, whether the authors checked the parental gene expression of these five circRNAs knockdown experiments? Considering the host genes of circRNAs normally serve their primary targets for gene expression.
3. The circRNA ASO showed the inconsistent regulation on host gene expression of HepG2 and Hep3B cell lines (Figures 9b and 9f), and the authors might propose the possible explanations for it.
4. In Figure 9i, there were five circRNAs delivering positively regulation on KPNA2 expression. It required the miRNA-specific reporter assays to confirm the circRNA-miRNA-mRNA interaction; further, the KPNA2 protein expression measured by Western blot assays was necessary for examination to support their hypothesis.
5. Continuing the question 4, whether the authors checked these five circRNAs were upregulated expressed in cancer tissues than normal sections? In the same time, the targeted miRNAs were hypothesized downregulated in tumor sections. Employing clinical data may provide supportive findings to consolidate their observations.
Minor
1. The primer sequences for miRNA-stem loop detection should be provided in the Method.
2. The colors and contrast of Figures 2-4 need to be improved for better interpretation.